# Cautious Bayesian Optimization: A Line Tracker Case Study

**DOI:** 10.3390/s23167266

**Published:** 2023-08-18

**Authors:** Vicent Girbés-Juan, Joaquín Moll, Antonio Sala, Leopoldo Armesto

**Affiliations:** 1Departament d’Enginyeria Electrònica (DIE), Universitat de València, 46100 Burjassot, Spain; vicent.girbes@uv.es; 2Instituto U. de Automática e Informática Industrial (ai^2^), Universitat Politècnica de Valencia, 46022 Valencia, Spain; josemollster@gmail.com (J.M.); asala@isa.upv.es (A.S.); 3Instituto de Diseño y Fabricación (IDF), Universitat Politècnica de Valencia, 46022 Valencia, Spain

**Keywords:** Bayesian optimization, safety constraints, experimental optimization, Gaussian processes, chance-constrained optimization

## Abstract

In this paper, a procedure for experimental optimization under safety constraints, to be denoted as constraint-aware Bayesian Optimization, is presented. The basic ingredients are a performance objective function and a constraint function; both of them will be modeled as Gaussian processes. We incorporate a prior model (transfer learning) used for the mean of the Gaussian processes, a semi-parametric Kernel, and acquisition function optimization under chance-constrained requirements. In this way, experimental fine-tuning of a performance objective under experiment-model mismatch can be safely carried out. The methodology is illustrated in a case study on a line-follower application in a CoppeliaSim environment.

## 1. Introduction

Learning by experiment is routinely carried out in many task optimization setups, either from scratch if no reliable model is available, or to fine-tune some controller parameters or setpoints in order to overcome process-model mismatch. Assuming a measurable performance index is available, several options do exist to carry out experimental optimization: direct (model-free) policy-search approaches with statistical gradient estimates [1,2]; extremum-seeking control paradigms [3]; identification based indirect optimization [4,5]; modifier adaptation, identifying a model error in the cost gradient [6,7], etc. Robust Control approaches may handle model/plant mismatch [8], but in such a case, data gathering will not improve performance. Improving performance in closed-loop control is dealt, in many cases, with adaptive control techniques [9]. Notwithstanding, closed-loop dynamics is out of the scope of the approach in this paper, i.e., the setup will discuss static function optimization, where decision variables will be set to fixed constants at start, so there will be no need of probabilistic state estimation or sensor fusion as in other real-time control applications [10,11] (even if there is a closed-loop line tracker control case study, we understand the problem as an “episodic” one, in which a scalar performance figure is obtained after each experiment; this is intentional in our problem statement). Thus, these aspects will not be considered any further. Learning by demonstration [12] may be another option in some classes of tasks in which examples can be provided, which is also out of the scope of this work.

In particular, plant–model mismatch can be characterized via a probabilistic model, giving rise to Bayesian Optimization (BO) [13,14]. The underlying probability model is usually a Gaussian Process (GP), which can predict unexplored values of a function in terms of mean and variance (marginal predictions are Gaussian, motivating the name). Gaussian Processes [14] are an interpolation/function approximator that offers an uncertainty bound compared to, let us say, function approximation using lookup tables [15]. The provided confidence intervals can guide exploration in experimental optimization and learning, which is why the GP approach is widely used in that context (Bayesian optimization). The BO idea has been applied to robotics policy search, see for instance [16], but also to other contexts, such as material science [17], environmental science [18], financial computations [19], medicine [20], etc. Bayesian optimization has also been used in fault diagnosis in engineering systems [21] or medical diagnosis [22].

Usually, the goal of BO is optimizing a given *static* “performance” function y=f(x); BO is commonly used when the actual function *f* is expensive to evaluate in some sense (economic or time resources), so a surrogate acquisition function is used as a proxy of it to decide the actual point in which *f* should be evaluated. Experimental optimization is usually such a case, in the sense that optimizing the acquisition function in the computer is usually far cheaper than carrying out actual experiments that take time, resources, and human intervention. In robotics application, we might have a simulation model available so tasks can be optimized on that model prior to actual costly experimental testing; these situation might be called “transfer learning”, usually accounted for in BO with a suitable prior on the mean and variance of f(x).

In some complex cases, especially when there is a finite maximum number of experimental samples, the BO problem needs considering the exploration–exploitation dilemma; in these cases, the optimizer would need looking ahead in a multi-step Bayesian optimization [23] setup reminiscent of predictive control. In fact, given that a probabilistic model is updated as data are gathered, the actual setup should be posed as a Partially observable Markov decision process (POMDP) [24]; a preliminary approach to (unconstrained) Bayesian optimization in the POMDP framework appears in the conference paper [25]. Most multi-step BO approaches, however, as pointed out by [26], seem to provide similar performances as conventional Bayesian optimization methods, and thus, given the marginal performance gain and the added computational burden reported in said reference, they are out of the scope of this paper.

In most cases, the operation point yielding optimal performance ends up being close to operational constraints (with some of them being active). If there is uncertainty in the model, there may also be uncertainty on the actual constraints. Bayesian optimization under constraints has been discussed in prior works, see the state-of-the-art review in [27,28]. In particular, in [29], a standard BO acquisition function was multiplied by the probability of not violating constraints to craft a new constraint-aware acquisition function; constraints may be violated if the expected improvement is high and the probability of violation is not high. However, in our case study, we have opted for a fixed probability level of constraint violation (5% chance constrained optimization, based on a confidence bound criterion). In [30], they pose the problem in a chance-constrained setting with expected value or expected improvement acquisition functions. In [28], a decoupled approach is considered where objective function and constraints can be separately evaluated and there is a limited budget on computational/time resources to be decided upon, based on Entropy Search. Nevertheless, in our problem setup, both constraints and performance objective function are evaluated in a coupled way. Hence, the considerations in the cited work are not relevant to our particular problem scope.

This paper presents a “constraint aware” Bayesian Optimization setting, inspired in [30], incorporating to the approach a model-based prior (transfer learning) and a semi-parametric Gaussian Process to describe the plant-model mismatch. The proposed ideas are developed in the context of a line-follower robotic application [31,32] to illustrate the main ingredients of the approach: computing a base prior model from simulation, detailing how to set up a relevant risk measure to our robotic application (a combination of low-frequency error plus high-frequency control activity), selecting a “sufficiently good” and “sufficiently safe” initial trial from the prior models, and building a semiparametric Gaussian process modeling of the simulation/experiment mismatch of performance objective and safety constraint functions.

The paper structure is as follows: Section 2 discusses the problem to be solved (constrained Bayesian optimization); Section 3 describes the main contribution of this paper, combining the above ingredients into a methodology to experimentally optimize a performance measure until safety-related constraints in robotic applications; Section 4 presents the results of the method applied to two case studies: one academic one-dimensional setup and another case study simulating a line-tracker robot; Section 5 closes the paper.

## 2. Problem Statement

Let us consider a static function f(x), which is (partially) unknown, which we must optimize while not violating some constraints. Note that in some application contexts, the actual performance *f* or safety *g* is a “summary” of a whole trajectory of a dynamic system; for instance, whether a control loop gets unstable or not, episodic robotics experiments tracking a given path, or determining whether a task has been accomplished, etc. Thus, we will use a static optimization setup, i.e., with no dynamics on *f* or *g*, but encoding “good performance” in just a number in some applications may require careful analysis. Our goal is progressively improving our performance f(x) (minimization) under constraints g(x)≤0 while acquiring knowledge on both *f* and *g* from samples. Given the safety constraints, experimental optimization must be adapted to a constraint-aware setup to obtain
$$x^{*}:=\arg\min_{g(x)\leq 0}f\left(x\right)$$

There may be more than one constraint, but we will concentrate on the single constraint case, as the ways to manage several of them will be analogous because all of them can be lumped onto a single function $g\left(x\right)=\max_{i}g_{i}\left(x\right)$. A chance-constrained optimization path will be pursued to avoid (with a given confidence/probability level) hitting invalid points g(x)>0.

The *k*-th experiment will provide an input $x_{k}$ to *f* and *g*, k=1,⋯, and it will obtain a sample of both $f_{k}:=f\left(x_{k}\right)$ and $g_{k}:=g\left(x_{k}\right)$. The problem will consider that no dynamics or time-variation is present in either *f* or *g*, so $f_{k}$ and $g_{k}$ will not depend on time or prior inputs. There may be a measurement noise with variance $\lambda_{f}$ and $\lambda_{g}$, respectively, in the obtained samples of *f* and *g*, i.e., replacing the above definition of $f_{k}$ with $f_{k}:=f\left(x_{k}\right)+\epsilon$, with ϵ being a random variable drawn from a zero-mean normal distribution of variance $\lambda_{f}$, and likewise for $g_{k}$.

The state of our optimization algorithm prior to taking decision $x_{k}$ will be the set of past actions $X_{k-1}:=\left\{x_{1},\cdots ,x_{k-1}\right\}$ and associated observations $F_{k-1}=\left\{f_{1},\cdots ,f_{k-1}\right\}$, $G_{k-1}=\left\{g_{1},\cdots ,g_{k-1}\right\}$. When the actual value of *k* is not relevant, subindices will be omitted for notational simplicity.

In summary, our problem can be stated as deciding the next experimental sample $x_{k}$ based on $X_{k-1}$, $F_{k-1}$ and $G_{k-1}$ to obtain a good performance value while keeping the risk of constraint violation below a given probability level (chance constrained optimization). The proposed solution to this problem will be addressed via suitable modifications to Bayesian Optimization techniques, to be discussed next.

## 3. Constraint-Aware Bayesian Optimization

This section will present how to adapt Bayesian optimization algorithms in the literature to incorporate constraints that should not be violated (in a chance-constrained setup, given the underlying probabilistic models).

The core of Bayesian Optimisation [33] is using past observations to craft an internal probabilistic model of *f* and *g* to help decide the next observation point $x_{k}$. In particular, such internal probabilistic models will be assumed to be Gaussian processes with known mean functions $\bar{f}\left(x\right)$ and $\bar{g}\left(x\right)$ given by an a priori model of the system under study, and known covariance kernel functions $\kappa_{f}\left(x_{1},x_{2}\right)$ and $\kappa_{g}\left(x_{1},x_{2}\right)$ encoding the smoothness of the underlying functions (the power spectral density is related to the Fourier transform of $\kappa_{f}$ or $\kappa_{g}$ in a stationary case). For later Bayesian optimization steps, it will be assumed that the “true” *f* and *g* are realizations of such Gaussian processes.

As mentioned in the introduction, BO methods [14] optimize a “surrogate” acquisition function to decide which will be the following sample $x_{k}$ actually to experiment; this is justified by the fact that computations on the acquisition functions are cheap compared to actual experiments. In addition to this, this search must be guided in a cautious way to avoid exploring areas forbidden by the constraints.

The main aspects of our proposal will be:How to craft the constraints in a way amenable to static Bayesian optimization;How to encode possible model-based knowledge on the problem (transfer learning);Setting up Gaussian processes and choosing the acquisition function;Incorporating possible offset/scaling biases via semi-parametric ingredients;Developing a final algorithm for experimental optimization, incorporating all the above.

### 3.1. Constraint Encoding

In a particular application, regarding constraints, there may be cases in which their definition is evident (such as operational limits). Still, there may be other cases where improving performance while avoiding failures requires carefully considering the definition of safety constraints.

In particular, in robotics and control tasks involving trajectories of dynamic systems, constraints must be an indicator of the “stability margin” or the “risk” of a given maneuver that summarizes a whole trajectory τ(x) in a number g(τ(x)), with *x* being the set of configuration parameters (decision variables) to optimize. There may be two approaches to the problem:*Direct safety-related measurements:* Minimum distance to obstacles, maximum accelerations close to operational limits, etc.;*Indirect safety-related measurements:* In some cases, failures are “catastrophic” but these measures are binary: say, a control loop is either stable or unstable, or the line-tracker robot in later section may lose track sight in sensors. A binary success/failure measure cannot be suitably handled with Gaussian processes without methodological modifications, as Gaussian process models are smooth functions that cannot model sharp discontinuities. Furthermore, just measuring “stable” does not tell you how close you are to instability, and, say, a slight modification of some decision parameters may give rise to failure. This is why indirect measurements are needed in these contexts. Thus, we can propose risk measures based on robust control ideas [34]:
–*Medium-to-high-frequency control action* components. The appearance of closed-loop resonances augments such medium-frequency activity in the recorded control signals. They indicate that the system’s Nyquist plot is approaching −1.–*Peak error and manipulated variable values*. Sensor and actuator saturation may cause the trajectory tracking or control problem to fail. Thus, maximum error or control action may also be monitored to compute *g*.An example of the above ideas will be discussed in the case study in Section 4.2.

### 3.2. Gaussian Processes and Bayesian Optimization

Given prior samples $D_{f}:=\left\{X,F\right\}$, a Gaussian process is a probabilistic model that incorporates these samples to compute a posterior GP prediction $\hat{f}\left(x\right)$ of the true f(x) at a point *x* (note that the dataset $D_{f}$ before decision *k* will be $\left\{X_{k-1},F_{k-1}\right\}$, but subindices have been removed to avoid notational clutter). The prediction is given by a normal distribution:(1)$$\hat{f}\left(x\right)∼N\left(\mu_{f}\left(x|D_{f}\right),\Sigma_{f}\left(x|D_{f}\right)\right)$$
with mean and variances given by:
(2a)$$\begin{array}{cc} \mu_{f}\left(x|D_{f}\right) & =\bar{f}\left(x\right)+\kappa_{f}\left(x,X\right){\left(\kappa_{f}\left(X,X\right)+\lambda I\right)}^{-1}\left(F-\bar{f}\left(X\right)\right) \end{array}$$
(2b)$$\begin{array}{cc} \Sigma_{f}\left(x|D_{f}\right) & =\kappa_{f}\left(x,x\right)-\kappa_{f}\left(x,X\right){\left(\kappa_{f}\left(X,X\right)+\lambda I\right)}^{-1}\kappa_{f}\left(X,x\right) \end{array}$$
where $\bar{f}\left(X\right)$ is a vector formed by evaluating $\bar{f}$ at each of the previous data samples and stacking them in column form, and an analogous abuse of notation is implicit in $\kappa_{f}\left(x,X\right)$ and $\kappa_{f}\left(X,X\right)$. Equivalent formulae will describe the posterior Gaussian estimate $\hat{g}\left(x\right)$ based on samples $D_{g}:=\left\{X,G\right\}$.

Bayesian optimization optimizes a surrogate acquisition function based on such posterior prediction instead of the actual (expensive) experimental *f*. The most-used options in the literature are [33]:Expected Improvement: average of a truncated Gaussian at the currently best value in $D_{f}$;Lower/Upper Confidence Bound (LCB, UCB): $\mu_{f}\left(x|D_{f}\right)\pm 2\sqrt{\Sigma_{f}\left(x|D_{f}\right)}$;Probability of Improvement: integral of the truncated Gaussian at the currently best value in $D_{f}$;Expected value: $\mu_{f}\left(x|D_{f}\right)$.

These acquisition functions strike a different balance between exploitation (which would amount to propose the *x* with minimum expected value, i.e., the last of the above options) and exploration (the rest of the options have some implicit exploratory component in them).

Furthermore, recent entropy-based acquisition functions have been proposed: the basic idea is determining the next test point based on the expected information gained about the location or the value of the optimum of a function; see [35] for details. As entropy search may involve nested Monte Carlo algorithms, the added complexity makes these options not popular in actual applications, given that the other above enumerated acquisition functions do yield good performance in most cases, so entropy-based acquisition functions will not be discussed further in this work.

### 3.3. Transfer Learning from Model to Experiment

In many applications, particularly robotic ones such as the case study to be later presented, we can easily build a first-principle model (kinematics, dynamics). With that model, we may solve optimization problems and run accurate simulations. However, real-life implementation will unavoidably have mass/inertia/friction mismatches. The first-principle model will capture the essential features of the application, but detailed fine-tuning steps cannot be captured with that prior model. Even if a Gaussian Process can, in theory, model the whole behavior of an actual experimental plant when fed with suitable data, it is only the model/reality mismatch the one needed to be considered in the BO setup, transferring the knowledge in the first-principle model to the GP as a non-zero mean function [36,37], also known as Bayesian model averaging. The usage of the prior first-principle model will allow to carry out BO with a lower amount of actual experimental data needed.

In this way, the (a priori) estimated mean of *f*, denoted as $\bar{f}$, would indicate a model-based performance estimation. The a priori mean of *g*, represented as $\bar{g}$, must be handled with care because if we wish to have a low probability of constraint violation, the model of *g* must also be provided with some confidence intervals. The advantage of transfer learning is that if the prior model is good enough, the initial exploration point will be close enough to the actual optimum so that the experimental optimization algorithms will start in the “basin of attraction” of the optimal point and will be less likely to be trapped in local minima than, say, other random initialization options.

The model-based setup will be used to guide iterations (via the mean of the GP for *f* and *g*), but its more important role will be determining the first experimental sample $x_{1}$. The first choice that comes to our mind would be:(3)$$x_{1}=\arg\min_{\bar{g}\left(x\right)\leq 0}\bar{f}\left(x\right)$$
which would be exploring *f* in the region where constraints are not violated according to the “mean” model $\bar{g}$. Note that the above constrained optimization in (Equation 3) is not carried out in the actual experimental plant but only in the computer, so it may be considered “cheap” in computational terms. Furthermore, there is no risk of violating constraints in the optimization iterations.

However, given that knowledge of *g* is assumed uncertain, the above search (Equation 3) may be too risky: if we believe $\bar{g}$ to be the mean of the “actual” *g*, then the said actual *g* would be below the mean 50% of the times; a 50% chance of constraint violation is usually unacceptable in most applications.

Then, a more cautious initialization is recommended, such as:(4)$$x_{1}=\arg\;\min_{\tilde{g}\left(x\right)\leq 0}\bar{f}\left(x\right)$$
where $\tilde{g}$ replaces $\bar{g}$ in (Equation 3), with $\tilde{g}$ being the a priori upper confidence bound of *g* before collecting any data, determined by the covariance matrix parameters of $\kappa_{g}$. If we chose a two-standard-deviation upper confidence bound:(5)$$\tilde{g}\left(x\right):=\bar{g}\left(x\right)+2\sqrt{\kappa_{g}\left(x,x\right)}$$
then we have only a 5% chance of constraint violation, according to textbook Gaussian formulae. That confidence level will be deemed good enough in our later case studies.

As a last option, the most cautious initialization would be $x_{1}=\arg\min\tilde{g}\left(x\right)$, disregarding performance for the initial sample and prioritizing avoidance of constraint violation. For each problem, the user must decide between the most risky option (first one) versus the most cautious one (last one); note that in quite a few practical industrial cases, the optimum performance point is close to the constraint limit, so these aspects may be of importance in actual applications.

#### Semi-Parametric Gaussian Process Models

If we have a reasonably accurate model, one frequent case of model/reality mismatch is an offset or scale error, say $f\left(x\right)\approx offset+scale\cdot \bar{f}\left(x\right)$. As another option, the said mismatch may be represented by a linear transformation of the model $\bar{f}$ or a set of regressors, such as $f\left(x\right)-\bar{f}\left(x\right)\approx \Phi \left(x\right)\theta$. However, in a general case, there may be an additional “residual” mismatch unable to be modeled by this parametric component.

Based on the above motivation, a Gaussian Process may be generated from a parameterized function approximator plus some random Gaussian process terms. In particular, we may consider:(6)$$f\left(x\right)=\bar{f}\left(x\right)+\Phi \left(x\right)\theta +\phi \left(x\right)$$
where $\bar{f}\left(x\right)$ is the assumed prior mean, $\theta \in R^{m}$ is a column vector of adjustable parameters sampled from a prior distribution with zero mean and covariance matrix $\Lambda_{\theta }$ (independent of *x*), Φ(x) is a row vector of known regressor functions, and ϕ(x) is a GP with zero mean and covariance kernel function $\kappa_{\phi }\left(x_{1},x_{2}\right)$. Expression (Equation 6) is known as a *semi-parametric* Gaussian Process model [38,39], which is a generalisation of the non-parametric GP component ϕ(x).

Under the above assumptions, the covariance kernel function of the GP generating *f* would be given by:(7)$$\kappa_{f}\left(x_{1},x_{2}\right)=\Phi \left(x\right)\Lambda_{\theta }\Phi {\left(x\right)}^{T}+\kappa_{\phi }\left(x_{1},x_{2}\right)$$
for later application of GP prediction/interpolation formulae ([Disp-formula FD2a-sensors-23-07266]) and ([Disp-formula FD2b-sensors-23-07266]), with mean function $\bar{f}\left(x\right)$. Note that the only requirement for converting ([Disp-formula FD2a-sensors-23-07266]) and ([Disp-formula FD2b-sensors-23-07266]) from non-parametric to semi-parametric version is using the modified covariance $\kappa_{f}$ in (Equation 7); actual values of the parameters can be recovered with suitable computations, omitted for brevity (see [38,39] for details).

Likewise, a similar semi-parametric description of the constraint function g(x) may be crafted, left to the reader for brevity.

The semi-parametric model is open to several interpretations:The prior knowledge on the plant should be encoded in $\bar{f}\left(x\right)$, perhaps incorporating some component $\Phi \left(x\right)\theta_{0}$ regarding “nominal” parameter values inside the said $\bar{f}$. Then, θ in the above GP model would encode parameter increments from the nominal ones.As another interpretation, if $\Phi \left(x\right)=\left[x^{T}\;\;1\right]$, the mismatch would be interpreted to be close to a linear function, which might locally be understood as uncertainty in a possible offset and uncertainty in the gradient of the model. Likewise $\Phi \left(x\right)=\left[x⊕x\;\;x^{T}\;\;1\right]$ would locally fit the error to a parabolic shape (x⊕x denotes the degree-2 monomials in *x* arranged in a row).Finally, if $\Phi \left(x\right)=[\bar{f}\left(x\right)\;\;1]$, the semi-parametric model would encode offset and scaling uncertainty, as the posterior mean should be close to:
(8)$$\left(1+\theta_{1}\right)\bar{f}\left(x\right)+\theta_{2}$$
and thus, $\theta_{1}$ will be denoted as the *scaling* parameter and $\theta_{2}$ will be referred to as the *offset* parameter.

Of course, combinations of several of the above options are possible if one wishes to indicate the structure of the uncertainty the experimental optimization phase must cope. Details are left to the reader, for brevity.

## 4. Case Studies

### 4.1. One-Dimensional Example

The first example of the proposed methodology will discuss a one-dimensional optimization problem in which we wish to optimize the cost function:(9)f(x)=0.8(tanh(3sin(x+1.2))−sin(x+1.7))+0.2
with the following safety constraint function:(10)$$g\left(x\right)=1.2{\left(x+1\right)}^{2}-4.15\leq 0$$

As usual in BO, the functions are assumed to be realizations of Gaussian processes; in this case, we chose the squared-exponential kernel:(11)$$\kappa_{f}\left(x_{1},x_{2}\right)=M_{f}e^{-\frac{{\left(x_{1}-x_{2}\right)}^{2}}{\sigma_{f}^{2}}}$$
(12)$$\kappa_{g}\left(x_{1},x_{2}\right)=M_{g}e^{-\frac{{\left(x_{1}-x_{2}\right)}^{2}}{\sigma_{g}^{2}}}$$
where $M_{f}=0.4$, $\sigma_{f}=0.2$ are the chosen kernel parameters of the cost function model f(x), and $M_{g}=10$, $\sigma_{g}=5$ are the chosen kernel parameters of the safety function g(x), for this particular example (we did carry out several tests and chose the presented kernel hyperparameter values by heuristic/trial-and-error; hyperparameter optimization was not considered in the case studies in this paper because, for the given examples, the number of samples is low and concentrated around the final solution except for a handful of initial samples—in these cases, such a hyperparameter search may be unreliable).

An offset plus scaling semiparametric component has also been incorporated from a prior model, considered to be the mean function:(13)$$\bar{f}=\tanh\left(3\sin\left(x\right)\right)-\sin\left(x+0.5\right)$$
and thus, $f\left(x\right)∼\bar{f}\left(x\right)+\theta_{1_{f}}\bar{f}\left(x\right)+\theta_{2_{f}}+\phi_{f}\left(x\right)$, with $\phi_{f}$ being the Gaussian process component of the cost function with the above covariance in (Equation 11). Note that we intentionally did set $\bar{f}$ to be a different function to f(x) in (Equation 9), to test how the GP approximates the mismatch. In most actual applications, the function in (Equation 9) is actually not known.

The same semiparametric model has been used for the safety function, where:(14)$$\bar{g}=x^{2}$$
which is the prior mean, so $g\left(x\right)∼\bar{g}\left(x\right)+\theta_{1_{g}}\bar{g}\left(x\right)+\theta_{2_{g}}+\phi_{g}\left(x\right)$, with $\phi_{g}$ being the Gaussian process component of the safety function with the above covariance in (Equation 12). Note that, intentionally, we have set $\bar{f}$ and $\bar{g}$ to be *different* to actual *f* and *g* in order to test transfer learning performance under model mismatch.

The prior mean of scaling parameters $\theta_{1_{f}}$ and $\theta_{1_{g}}$, as well as offset parameters $\theta_{2_{f}}$ and $\theta_{2_{g}}$, are set to zero, whereas the prior standard deviations for the scaling parameters are set to 0.2 and the prior standard deviation for the offset parameter $\theta_{2_{f}}$ is set to 0.2 and for $\theta_{2_{g}}$ to 2.

In our implementation of Algorithm 1 below, we opted to use Matlab Optimization Toolbox 2022 using the fmincon function.
**Algorithm 1:** Final algorithm proposed for constraint-aware Bayesian optimizationSet k=1. Obtain an initial sample $x_{1}$ by optimizing the a priori transfer-learning model $x_{1}:=\arg\min_{x}\bar{f}\left(x\right)$ subject to $\tilde{g}\left(x\right)\leq 0$, where $\tilde{g}\left(x\right)$ is the initial prior UCB of *g*. If we prioritize risk avoidance, the initial sample might be $\min_{x}\tilde{g}\left(x\right)$.Carry out the actual experiment with $x_{k}$.Incorporate $x_{k}$ to the sample Thete $f(x_{k})$ and $g(x_{k})$ to the records $F_{k}$ and $G_{k}$, respectively, and rebuild the GP conditional predictions incorporating these new data.Set $f^{*}=\min_{k}f\left(x_{k}\right)$, set $x^{*}=\arg\min_{k}f\left(x_{k}\right)$, and set k=k+1.Obtain sample $x_{k}$ as $x_{k}=\min_{x}\tilde{f}\left(x\right)$ subject to $\tilde{g}\left(x\right)\leq 0$, being $\tilde{f}\left(x\right)$ a chosen acquisition function for BO of *f* and $\tilde{g}\left(x\right)$ the upper confidence bound with confidence set to 1−ε, with a sufficiently small ε>0 (chance constrained, risk level (for instance, replacing the prior variance by the posterior one in (Equation 5), we would have $\tilde{g}\left(x\right):=\bar{g}\left(x\right)+2\sqrt{\Sigma_{g}\left(x|D_{g}\right)}$ if we wished ϵ=0.05.)).Repeat from step 2 until maximum number of samples is reached or a chosen progress measure in $f^{*}$ stops improving.Return $x^{*}$ and $f(x^{*})$ as a result of the safety-constraint aware BO.

Figure 1 shows the progress of the constraint-aware Bayesian optimization iterations, starting from x=0 (labeled as sample “1”), using the lower confidence bound as the acquisition function, constrained to the upper confidence bound of *g* being negative. We can see that samples converge to the minimum, and safe operation constraints are not violated in any of the iterations.

### 4.2. Robot Line Follower

In this second case study, we have prepared a simulated environment combining CoppeliaSim^®^ and Matlab^®^ to validate the proposed method on a line follower robot; we chose the line follower, as it is a popular low-cost robotic platform [31,32], with a well-known behavior so intuitive conclusions can be quickly drawn to understand the performance of our proposals. Two circuits for a line-follower robot have been used to test the transfer-learning features, one acting as a “training” setup for the “model” initialization, the second acting as a “test” rig for Bayesian optimization under model mismatch.

The objective of the optimization task is determining optimal parameters (forward speed and proportional controller gain) to complete a lap in the circuit in minimum time without taking excessive risk of getting off the track (failure). Let us detail the different aspects involved in this setup.

#### 4.2.1. Simulation Environment

Matlab controls the execution of CoppeliaSim by performing simulations of the robot starting on the same initial conditions but with different controller parameters. Data are collected and returned to Matlab during each simulation to obtain the specific experiment’s performance and risk index. The constrained-aware Bayesian optimization method includes collected data to update its posterior belief of the performance and constraints and provides a new set of control parameters to simulate, as discussed in previous sections of this paper. The procedure is repeated until convergence or a maximum number of iterations has been reached.

Matlab controls the simulation synchronously so that every 10 ms, we can execute a simulation step in CoppeliaSim and update the robot state and sensor readings. Thus, the controller and simulation sampling times are the mentioned 10 ms.

The simulation reproduces the behavior of a differential configuration robot that uses low-cost electronics based on two DC motors for driving the robot base and three infrared sensors TCRT-5000. TCRT-5000 infrared sensors provide an analog output that varies based on the light intensity reflected by an infrared transmitted due to surface color changes or object detection. The robot controller uses the sensor in the middle (see Figure 2) to follow the edge of a black line on the ground using its analog output. The electronics of a TCRT-5000 sensor also includes an analog comparator to provide a digital output to detect the presence (or absence) of an object, or in this case, the line. This aspect is used to detect a mark on the circuit to detect the end of it, using the other two infrared sensors.

DC motors have been modeled in CoppeliaSim as torque joints configured in velocity mode. The simulation also uses a conventional differential drive configuration for wheeled mobile robots (see [40] as an example), where the caster wheel uses two torque-free joints to reproduce a contact wheel with low friction (i.e., rolling and orientation of this wheel is controlled by Bullet 2.78 physics engine in CoppeliaSim as part of the dynamic behavior of the simulation).

On the other hand, to reproduce measurements with TCRT-5000 sensors, we have simulated them with three vision sensors in the CoppeliaSim component library: the one in the middle uses a resolution of 32 × 32 pixels (we compute the mean of all pixels of the grayscale image, where 0 means black and 1 means white and return the mean value as if they were normalized analog measurements), while the other two sensors use a 1 × 1 vision sensor (image is binarized using a 0.15 threshold value returning true if the track-end mark is being detected).

When the simulation is running, CoppeliaSim simulates sensor readings and implements a control law based on:(15)$$\begin{array}{cc} \omega & =-k\cdot e \end{array}$$(16)$$\begin{array}{cc} \omega_{L} & =\frac{1}{R}\left(v-b\omega \right) \end{array}$$(17)$$\begin{array}{cc} \omega_{R} & =\frac{1}{R}\left(v+b\omega \right) \end{array}$$
where *e* is the error, i.e., the difference between the expected measured value at the edge of the line and the actual measurement, *k* is a proportional gain to regulate the angular correction, $\omega_{L}$ and $\omega_{R}$ are the reference angular velocities applied to the robot’s joints, and *R* and *b* are wheel radius and wheel separation distance (to the robot’s center), respectively.

The robot parameters are R=0.03 m and b=0.1 m and the range of *k* and *v* we are going to explore to optimize the controller are k∈[1,80] and v∈[0.1,1.2] m/s.

#### 4.2.2. Performance and Safety Measures

From a record of simulation trajectory data τ(x) (sensor readings, control action commands), we need to craft a suitable objective function f(x) and a safety constraint function g(x) for a circuit. As discussed above, our decision vector *x* will be two-dimensional, i.e., x≡(k,v).

**Performance measure.** The time taken to complete one lap of the circuit will be considered the performance to optimize. As small speeds yield large completion times, the natural logarithm of that lap time is the chosen performance figure for computation and plots as logarithm is a monotonic function. If a maximum time has been reached without completing the lap or sensors have lost the track image, then a large maximum penalty value is recorded. In contrast to the previous case study, it is not possible to have any analytical representation of f(x) in this second case: we can just run a simulation/experiment and see what it yields (numerically) after each episode.

**Safety constraint measure.** Regarding safety requirements, basically, we wish to complete a lap in the fastest possible way as long as track image is not lost from the sensors viewing range, and unmodeled dynamics or nonlinearity does not make the proportional closed-loop control unstable. Intuitively, excess speed *v* and excess feedback gain *k* might be dangerous, as well as too low a feedback gain because in this case the line tracker will not react energetically enough in curves.

The issue, of course, is that we do not know which are those dangerous speed and gain ranges, so we need “indirect” measurements to ascertain the risk of a given maneuver, as discussed in the main section. In particular, the “aggresiveness” of the control will be measured by a high-pass filtering of the control action commands, and then computing the RMS value of that high-pass signal. Furthermore, the “lack of good control” will be measured by the RMS value of the error signal E.

The high-pass filter will be a first order on:(18)$$H\left(z\right)=\frac{\alpha (z-1)}{z-\alpha }$$
associated with the difference equation:(19)$$u_{k}^{f}=\alpha u_{k-1}^{f}+\alpha \left(u_{k}-u_{k-1}\right)$$
where $u_{k}^{f}$ is the output of the filter (filtered control action) and the actually chosen value of α is 0.25, which entails that the cutoff frequency is $\omega_{c}=-\log\left(\alpha \right)/Ts=139$ rad/s, i.e., 22.1 Hz. Figure 3 depicts the Bode diagram of the chosen input filter.

Thus, the overall safety function will be:(20)$$g\left(k,v\right)=\ln(10^{3}rms{\left(E\right)}^{2}+0.5\cdot rms\left(u^{f}\right))-1.8$$
where 1.8 is a safety threshold value defining what we understand as the maximum acceptable risk level.

#### 4.2.3. Overall BO Setup

As we discussed, we simulate one robot and circuit, to gather a first set of values of *f* and *g*, which will be used as the mean (prior knowledge $\bar{f}$, $\bar{g}$ for transfer learning) for the Gaussian Processes in Bayesian Optimization. Simulations have been carried out on a dense enough grid of values and the 2D lookup table output will be used as $\bar{f}$ and $\bar{g}$.

We then optimize on a different circuit and with slight parameter changes on the robot, simulating what would be the “costly” experimental tests with *f* and *g* differing from the $\bar{f}$ and $\bar{g}$ obtained with the first circuit. Figure 4 conceptually depicts the above scheme.

The actual training and test functions *f* and *g* as well as the safety threshold are represented in Figure 5 below. In particular, the figure presents the actual functions *f* and *g* from the training (left column) and testing (right column) circuits. First row depicts the performance function *f* to be minimized, second row shows the safety function. The safety threshold (maximum admissible “risk”) is presented as a light green contour: of course, it is an actual contour of *g* in second row, but superimposed to *f* it gives us which will be the maximum performance attainable given the set risk limits.

The Gaussian processes modeling our performance and constraint functions will use the squared-exponential kernel, with $M_{f}=0.05$ and $\sigma_{f}=\left[20\;0.6\right]$ for the cost function, and $M_{g}=0.14$ and $\sigma_{g}=\left[8\;0.2\right]$ for the safety function, chosen by trial-and-error as discussed in Section 4.1. Measurement noise with standard deviations $\sigma_{noise_{f}}=0.04$ and $\sigma_{noise_{g}}=0.06$ have been added to the samples, setting $\lambda =\sigma_{noise_{f}}^{2}$ in ([Disp-formula FD2a-sensors-23-07266]) and ([Disp-formula FD2b-sensors-23-07266]).

Semiparametric components have been added, following (Equation 8), with zero prior mean. Scaling parameter will have a standard deviation of 0.2 for the models of *f* and *g*; the offset parameter will have standard deviations of 0.9 and 1.2 for the models of *f* and *g*, respectively.

#### 4.2.4. Line-Follower Results

In this section, we present some figures of the results of the proposed approach. The progress of the iterations on the test circuit is shown in Figure 6. We see how performance improves over iterations and how initial “cautious” decisions (large safety margin, low *g*) are taken and, once knowledge on *f* and *g* is gathered, iterations proceed to riskier decisions but stopping at the zero limit to *g*, as defined in our constraint in the problem statement.

As the ability to explore is determined by how our estimate of the safe region is built, the resulting estimated of *g* at the last iteration is presented in Figure 7. The mean estimate is the green colormap between the two dark blue surfaces representing the upper and lower confidence bounds (where the true safety boundary is within those bounds, as shown in Figure 8). The red plane illustrates the zero threshold determining the maximum acceptable risk. Then, the safe exploration region will be, with a 95% chance of being safe, the intersection of said plane with the upper confidence bound, which is depicted with a solid red line. Note that the distance between the upper and lower confidence bounds reduces to the measurement noise standard deviation in points that have actually been tested.

The last figure presenting results is Figure 8. In the figure, we present again the top-right plot in Figure 5 (i.e., the “test circuit” actual performance as a function of decision variables, plus the light-green contour delimiting the admissible exploration region due to the safety constraint *g*). In the new figure, we added as gray squares the sampled values of *k* and *v* in the iterations, progressing in the downhill direction as depicted in Figure 6a. The upper confidence bound of *g* estimated from these samples, when incorporating them into the Gaussian Process modeling *g*, intersects with the zero-level plane at the red contour.

Results show that the maximum performance level within the safe constraint limits has been reached.

## 5. Conclusions

In this paper, we have presented a methodology for constraint-aware Bayesian optimization in performance improvement tasks. Two Gaussian processes are used and updated as samples become available, one for performance estimation and the other for constraint estimation. The methodology incorporates transfer learning by using prior knowledge gathered by model-based computations (or simulations in a training circuit, in our case study) as the mean of these Gaussian processes. The performance optimization is carried out only on the set of candidate decision variable values, in which the upper confidence bound of the constraints is below a given threshold, indicating the maximum “risk” level to be permissible for the application under scrutiny.

The results for the one-dimensional academic example and for a CoppeliaSim-simulated environment for a line-tracking robot show the potential of the presented approach.

## Figures and Tables

**Figure 1 sensors-23-07266-f001:**
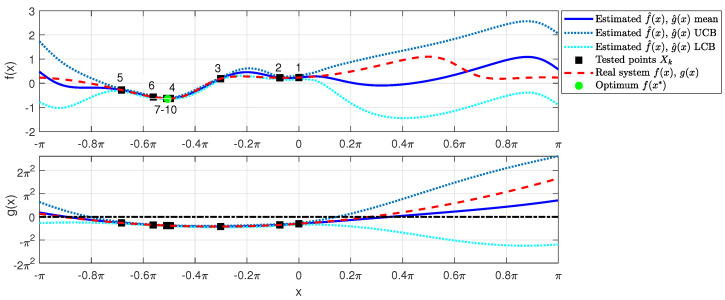
Estimated cost function $\hat{f}\left(x\right)$ and safety function $\hat{g}\left(x\right)$ (in solid blue, representing the mean) and real cost and safety functions (in dashed red). The estimated cost function’s upper and lower confidence bounds are shown in dotted light blue/cyan colors. The black squares are the tested values $x_{k}$, and the green bullet is the safe minimum value of the estimated cost function. The dashed-dotted black line highlights the zero threshold of the safety constraint.

**Figure 2 sensors-23-07266-f002:**
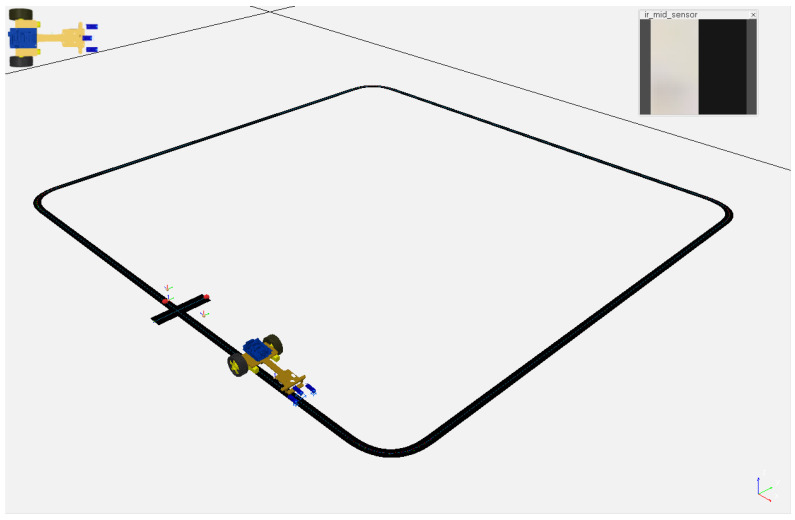
Line tracker in CoppeliaSim robot simulator.

**Figure 3 sensors-23-07266-f003:**
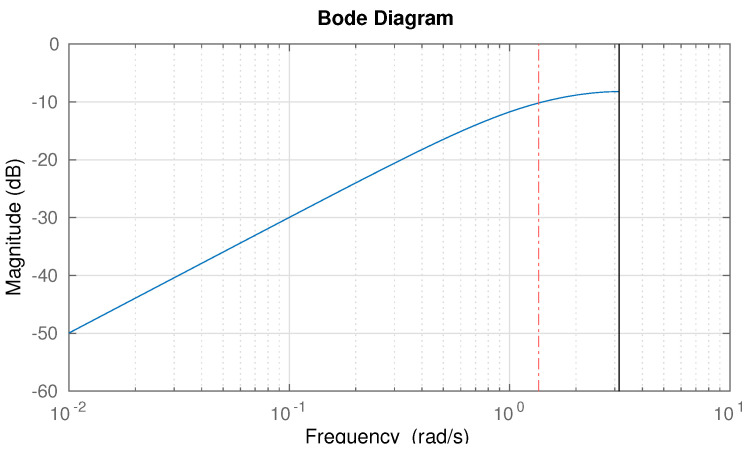
Bode magnitude plot of the control-effort high-pass weight for safety constraint evaluation.

**Figure 4 sensors-23-07266-f004:**
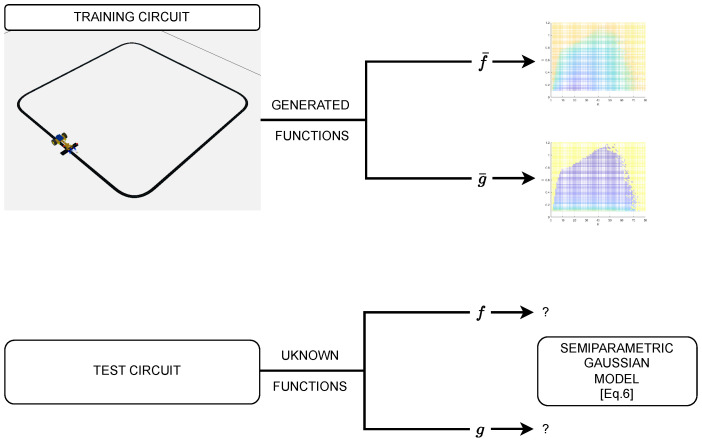
Summary scheme of the two-circuit CoppeliaSim setup.

**Figure 5 sensors-23-07266-f005:**
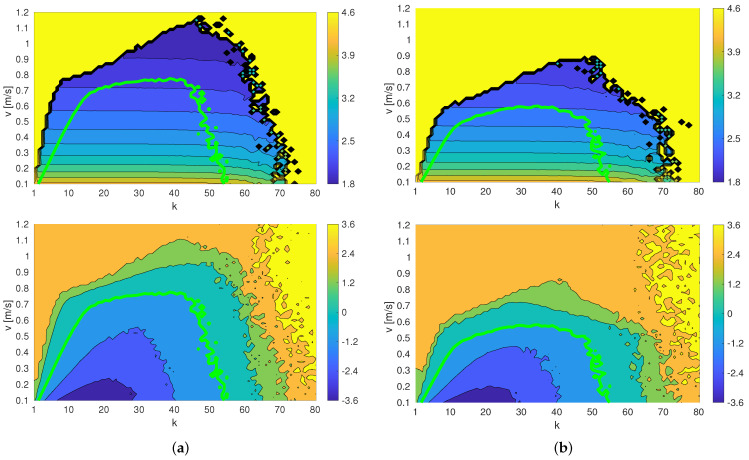
Cost function f(x) and safety function g(x): (**a**) training circuit and (**b**) testing circuit. The green lines determine the safety limits of each circuit (it is a threshold of *g*, but it has been overlayed onto *f* to highlight the admissible search region. The top row shows the plots for *f*, while the bottom row shows the plots for *g*.

**Figure 6 sensors-23-07266-f006:**
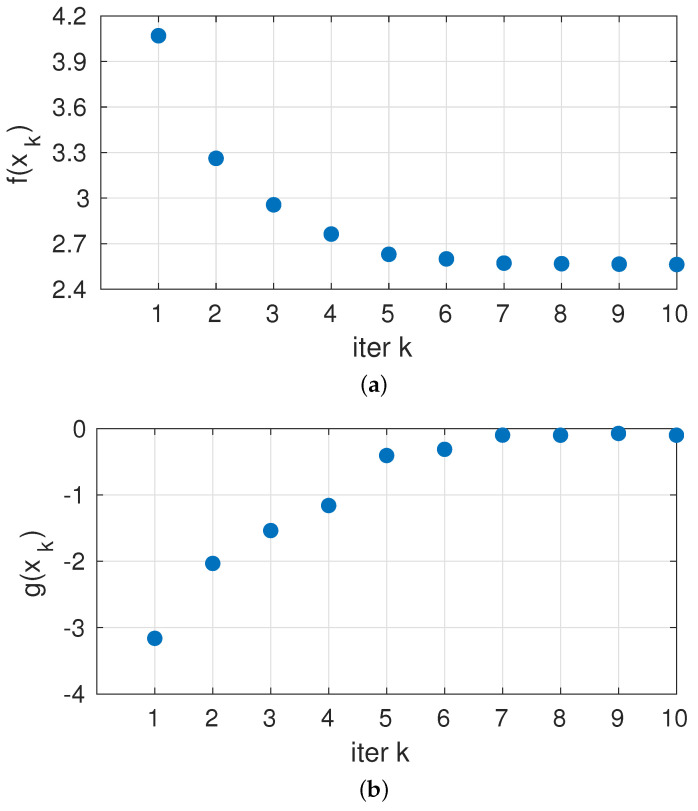
Values of (**a**) Cost function and (**b**) Safety function against test iterations.

**Figure 7 sensors-23-07266-f007:**
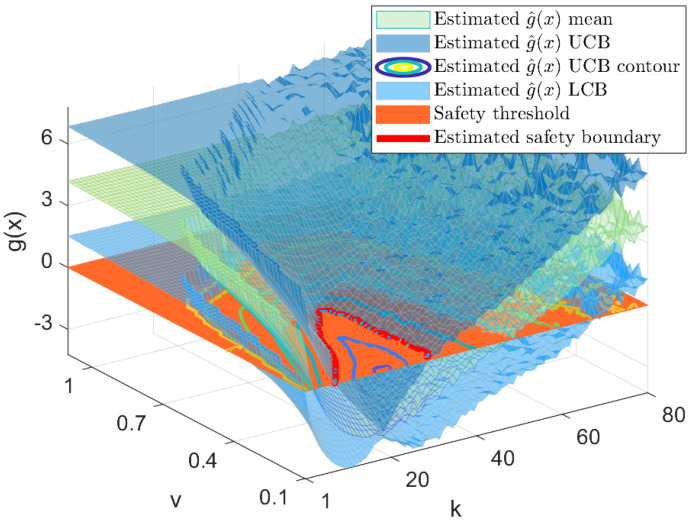
Estimate of the safety constraint $\hat{g}\left(x\right)$ from experimental samples at the last iteration: the mean of the constraint is depicted with a green mesh, UCB and LCB are depicted with blue meshes, the orange plane depicts the zero threshold level, and the red contour line determines the estimated safety boundary (other contour lines of the UCB are also shown).

**Figure 8 sensors-23-07266-f008:**
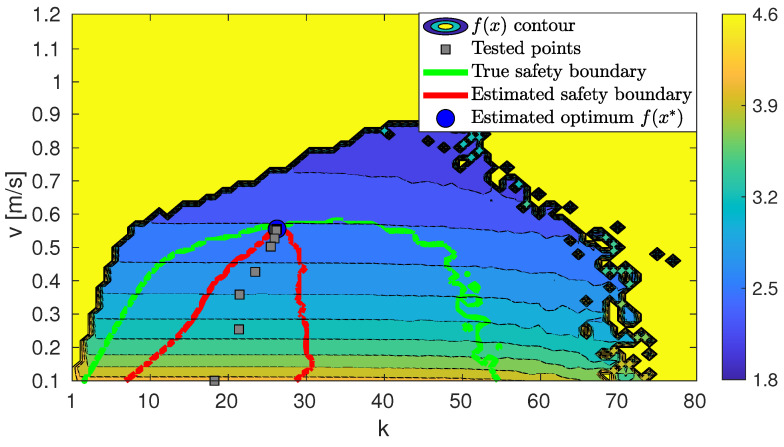
Test circuit f(k,v), with estimated (red) and actual (green) safe regions.

## Data Availability

No new data were created or analyzed in this study. Data sharing is not applicable to this article.

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
