# Peer review of "Cautious Bayesian Optimization: A Line Tracker Case Study"

_sensors, 2023, doi:10.3390/s23167266_

Round 1

Reviewer 1 Report

See attached file.

Reviewer 2 Report

The authors could have explained the problem through a real-world example to invigorate the readers' interest. The problem identified and addressed are relevant to the AI community.. The proposed method is proven to be an invaluable one through a simulation environment. The leverage of transfer learning to simplify and optimize smaller models is a welcome step. The constraint-aware Bayesian optimization algorithm articulated in the paper is easy to understand. 

For substantiating the proposed methodology, the authors have discussed a one-dimensional optimization problem in which they have optimized the cost function.  Please discuss about a multi-dimensional optimization case study also. 

The simulation tool used and the associated details are well-explained. The paper readability has to be improved a bit

The authors have proposed a new procedure for constraint-aware Bayesian Optimization. The research contributions of this paper are good and hence recommended for publication. The language part needs a moderate relook. The case study chosen for substantiating their new methodology is not sufficient. Especially a multi-dimensional optimization problem has to be pinpointed along with the details of multiple functions and solved through their newly exposed procedure

Reviewer 3 Report

Please find my comments in the attached document.

The paper is well written. 

Reviewer 4 Report

In this paper, experimental optimization under security constraints is regarded as a Gaussian process, combined with a prior model (transfer learning) to optimize the mean value of the Gaussian process, and fine-tune the objective function of the mismatch between the experiment and the model. As the results are not well discussed and more data should be provided, the paper should have a major revision. Some point-to-point suggestions are listed as follows:

1. In lines 21 and 56 of page 1 and 2, the authors describes that closed-loop dynamics, marginal performance benefits and computational burden are not considered in this paper. Could you explain in detail why these issues are not considered, or if the applicability of the method could be guaranteed in this way?

2. In paragraph 2 (46 lines) on page 2, the author simply describes the possibility of incorporating prior models (transfer learning) into Bayesian optimization. Related works  be reviewed as it is a key idea of the study.

3. In Section 4.1 of page 7 and 8, when x is less than 0 in Figure 1, the test points conform to the estimated mean of f(x) and g(x), but there is no discussion about the differences between the real system f(x) and g(x), as well as the estimated results when x is greater than 0. 

4. The performance of the method in this study should be compared with the method in reference [32].

5. On page 10, section 4.2.2, the robot's motion objective function f(x) should be provided.

Moderate editing of English language required

Round 2

Reviewer 3 Report

I am satisfied with the authors response. Thank you.

The paper is well written. 

Reviewer 4 Report

The paper has been well modified. It can be published at this stage.